# Pharmacological interventions for addressing pediatric and adolescent obesity: A systematic review and network meta-analysis

**Shuo Yang** [1☯], **Shuangqing Xin** [1☯], **Ronghui Ju** [2‡*], **Peizhuo Zang** [3‡*]

**1** Department of Endocrinology and Metabolism, The People's Hospital of China Medical University, The People's Hospital of Liaoning Province, Shenyang, China, **2** Department of Radiology, The People's Hospital of China Medical University, The People's Hospital of Liaoning Province, Shenyang, China, **3** Department of Cerebrovascular Disease Treatment Center, The People's Hospital of China Medical University, The People's Hospital of Liaoning Province, Shenyang, China

☯ These authors contributed equally to the study and are joint first authors.
‡ RJ and PZ also contributed equally to the study and are joint corresponding authors.
* d10@lnph.com (PZ); d8299@lnph.com (RJ)

## Abstract

### Background

Obesity significantly impacts the health outcomes of children and adolescents, necessitating a comprehensive study to evaluate the effects of various anti-obesity medications (AOMs) on weight-related and metabolic outcomes.

### Methods

PubMed, EMBASE, and CENTRAL were searched for studies published up to January 3, 2024. We performed a network meta-analysis on randomized clinical trials that compared various treatments for pediatric and adolescent obesity, such as phentermine/topiramate, semaglutide, exenatide, liraglutide, topiramate, metformin, fluoxetine, metformin/fluoxetine, sibutramine, and orlistat. The study evaluated body mass index (BMI), BMI percentage change, weight, BMI-SDS, waist circumference, metabolic, anthropometric, and safety outcomes.

### Results

The study gathered 2733 studies, including 30 articles that involved 3822 participants. The results of our research showed that PHEN/TPM was better at lowering BMI than exenatide, liraglutide, metformin, fluoxetine, Met/Flu, topiramate, orlistat, and sibutramine, with mean differences (MD) ranging from −10.29 to −1.28. Additionally, semaglutide demonstrated superior efficacy over other AOMs (MD ranged from −8.28 to −1.24). Various levels of certainty, ranging from very low to moderate, supported the findings. Furthermore, semaglutide demonstrated superior efficacy over exenatide (MD-12.43, 95% CI −23.95 to −0.30) regarding percentage change in BMI. Semaglutide also showed enhanced

**Data availability statement:** All relevant data are within the manuscript and its Supporting Information files.

**Funding:** The author(s) received no specific funding for this work.

**Competing interests:** The authors have declared that no competing interests exist.

weight reduction effectiveness compared to seven other AOMs except for PHEN/TPM (MD ranging from –15.56 to –12.65). Similarly, PHEN/TPM displayed greater weight reduction effectiveness than seven other AOMs, except for semaglutide (MD ranged from –12.17 to –9.27). Moreover, semaglutide proved more effective in decreasing waist circumference when compared with other AOMs apart from PHEN/TPM (MD ranged from –11.61 to –6.07). Similarly, we found that PHEN/TPM, excluding semaglutide and sibutramine, was more effective in reducing waist circumference (MD ranged from –8.64 to –5.51).

## Conclusions

The study found that semaglutide outperformed other AOMs in reducing BMI and additional weight-related outcomes in children and adolescents with obesity, while PHEN/TPM showed comparable efficacy.

## 1. Introduction

Obesity challenges public health in all age groups, particularly children and adolescents. Excessive body fat, or obesity, is a complicated, multifaceted chronic illness that can harm a person's health. Experts project that the global number of overweight and obese children and adolescents will reach 294.1 million by 2030 [1]. Children who are overweight or obese may experience health problems such as dysglycemia, hypertension, dyslipidemia, nonalcoholic fatty liver disease, and obstructive sleep apnea, as well as the development of cardiovascular disorders and various cancers, all while experiencing compromised mental health and a diminished quality of life [1–3]. As a result, childhood obesity has profound implications and poses a threat to an adult's health.

In the absence of intervention, childhood and teenage obesity are likely to persist into adulthood [4]. It can be challenging for adults, children, and adolescents to lose and keep weight off. Anti-obesity medicine (AOM) is an option for obese kids whose lifestyle therapies fail, although it is not widely available [5,6]. For teenagers, the Food and Drug Administration has approved phentermine, liraglutide, orlistat, semaglutide, and phentermine–topiramate. The American Academy of Pediatrics (AAP) released clinical practice guidelines (CPG) in 2023 that support the immediate use of AOM in adolescents aged 12 and above who have obesity, regardless of the presence or absence of other medical conditions [7,8]. However, starting AOM in clinical practice depends on several factors, including the patient's age and BMI, the presence of obesity-related comorbidities, the response to lifestyle therapy, and patient or family preferences. Considering that the weight loss efficacy and metabolic effects of various AOMs may vary, we used a Bayesian inferential framework to look at all the evidence from all the randomized controlled trials (RCTs) to compare the efficacy of different AOMs for treating obesity in children and adolescents.

## 2. Methods

This network meta-analysis has been reported following the Preferred Reporting Items for Systematic Reviews and Meta-Analyses (PRISMA-NMA) [9]. All research was conducted according to a protocol recorded in the PROSPERO database (CRD42024502250).

### 2.1. Search strategy

We comprehensively searched PubMed, Embase, and the Cochrane Library Central Register of Controlled Trials (CENTRAL) from its creation to January 3, 2024. Appendix 1 in S1

File provides a detailed description of the search plan. To find further studies, we manually searched ClinicalTrials.gov, pertinent meta-analyses, and the references of the included trials.

## 2.2. Study selection

Trials were eligible if they (1) enrollment of pediatric and adolescent individuals (with a mean age below 18 years at the onset of the intervention) diagnosed with obesity; (2) comparison of pharmacological interventions (including Metformin, Albiglutide, Dulaglutide, Semaglutide, Liraglutide, Lixisenatide, Taspoglutide, Exenatide, Sibutramine, Orlistat, Fluoxetine, Rimonabant, lorcaserin, Diethylpropion, Mazindol, Phentermine, Topiramate) among themselves or with a control group (defined as placebo or no treatment); (3) had a follow-up of at least 12 weeks; (4) provision of data regarding any of the predetermined endpoints; (5) inclusion of randomized clinical trials; and (6) publication in the English language.

We excluded studies with a history of prior bariatric surgery, uncontrolled thyroid disorder, secondary aetiologies for obesity, diabetes mellitus, severe depressive illness during the two years before the screening, and a documented history of a suicide attempt.

## 2.3. Data extraction and quality assessment

Two reviewers, S.Y. and RH.J., separately extracted data using a pre-established form based on relevant Cochrane Handbook for Systematic Reviews templates [10]. PZ.Z., the third reviewer, settled conflicts through discussion or agreement. We extracted both study and patient characteristics, which included first author, publication year, national clinical trial (NCT) number, study phase, study drug and control treatments, length of follow-up, background treatments, inclusion criteria, mean age, race, proportion of men, weight, height, BMI, and waist circumference. We would only use the most comprehensive and/or recently reported data if we obtained multiple reports from the same demographic. Only information from randomized controlled periods was utilized for studies with open-label extension periods. We can add multiple pairwise comparisons from a single study to the network without duplicating data, and when applicable, we included studies with multiple treatment groups as multi-arm studies. In cases where the research lacked sufficient data, the reviewers contacted the primary authors to gather and confirm the material. We attempted to contact the primary authors of four articles to request further study data but have yet to receive a response.

Following training and calibration exercises, reviewers used the Cochrane risk-of-bias tool for randomized trials (RoB 2.0) [11] for each eligible trial, evaluating five domains: the randomization process, deviations from intended interventions, missing outcome data, measurement of the outcome, and selection of the reported result. Studies were graded as (1) "low risk of bias" when a low risk of bias was determined for all domains; (2) "some concerns" when at least one domain was assessed as raising some concerns but not as being at high risk of bias for any single domain; or (3) "high risk of bias" when a high risk of bias was determined for at least one domain or the study judgment included some concerns in multiple domains. Two reviewers separately assessed the risk of bias (SQ.X. and RH.J.). In case of a disagreement, we consulted a third reviewer, PZ.Z. If we included ten trials or more for a given outcome, we used a comparison-adjusted funnel plot to identify small study effects.

## 2.4. Outcomes

The primary outcomes were body mass index (BMI) and BMI percentage change. Additional weight-related outcomes included weight, BMI standard deviation score (BMI-SDS) changes, and waist circumference. Metabolic outcomes (cholesterol, triglycerides, fasting blood glucose, fasting insulin, and the homeostasis model assessment insulin resistance index (HOMA-IR)).

We also examined blood pressure, heart rate, and safety outcomes. The three key safety outcomes of interest were gastrointestinal disorders, depression, and serious adverse events.

### 2.5. Data synthesis and statistical analysis

Network meta-analysis enables direct and indirect comparisons of several interventions, even when direct trial evidence is sparse. The researchers employed the odds ratio (OR) and weighted mean difference (WMD) to compare dichotomous and continuous variables. Each outcome had 95% confidence intervals (CIs) provided.

A direct meta-analysis was carried out using the DerSimonian and Laird random-effects models. We performed a Bayesian hierarchical network meta-analysis. We selected fixed or random effects models for each outcome based on the deviance information criterion (DIC). If the difference was more significant than 5, we used the model with the smallest value (Appendix 2 in S1 File). For Bayesian analysis, we used the Markov Chain Monte Carlo method for the estimation with 50,000 iterations, including a 10,000-iteration burn-in period and a 1-iteration thinning interval. We assessed statistical heterogeneity using the $I^2$ statistic, where values over 50% indicate substantial heterogeneity. We ranked the treatments using the surface under the cumulative ranking curve (SUCRA) and mean ranks for each outcome. We planned to use a design-by-treatment test [12] to check the assumption of consistency across the entire analytical network and the node-splitting method to evaluate the local inconsistency of the model by separating evidence for a particular comparison into direct and indirect evidence. However, due to the lack of closing loops, direct estimates could not be obtained in this review.

We performed no subgroup or sensitivity analysis due to the sparse data, except for the primary outcome, BMI. We subgrouped the outcomes by BMI ($\geq 35\,\mathrm{kg/m^2}$ or $< 35\,\mathrm{kg/m^2}$) to assess any differences in effect within these subgroups. We performed the sensitivity analysis by excluding open-label trials. We also performed network meta-regressions on BMI at baseline, age, and follow-up times on the primary outcomes of BMI to better understand the influence of patient characteristics on drug efficacy.

The GeMTC package on R (version 3.6.3) will be used for statistical analysis on network meta-analysis, assessment of global heterogeneity, and network meta-regression; STATA 14.0 software will be used for pairwise meta-analysis, estimation of transitivity, and funnel plots.

### 2.6. Summary of findings and assessment of the certainty of the evidence

We planned to use the CINeMA system (Confidence in Network Meta-Analysis), an adaptation of the GRADE (Grading of Recommendations, Assessment, Development, and Evaluations) approach, to assess the evidence for network meta-analysis [13,14]. The CINeMA is a web application that simplifies the evaluation of confidence in network meta-analysis findings by considering six domains: within-study bias, reporting bias, indirectness, imprecision, heterogeneity, and incoherence.

## 3. Results

### 3.1. Study characteristics

Fig 1 shows a flow chart for trial selection. We retrieved 2733 studies, of which 30 articles enrolling 3822 participants were included in the network meta-analysis (Appendix 14 in S1 File for a complete reference list). Appendix 3 in S1 File shows the characteristics of the studies. We analyzed eleven treatments, which included phentermine/topiramate (PHEN/TPM), semaglutide, exenatide, liraglutide, topiramate, metformin, fluoxetine, metformin/fluoxetine

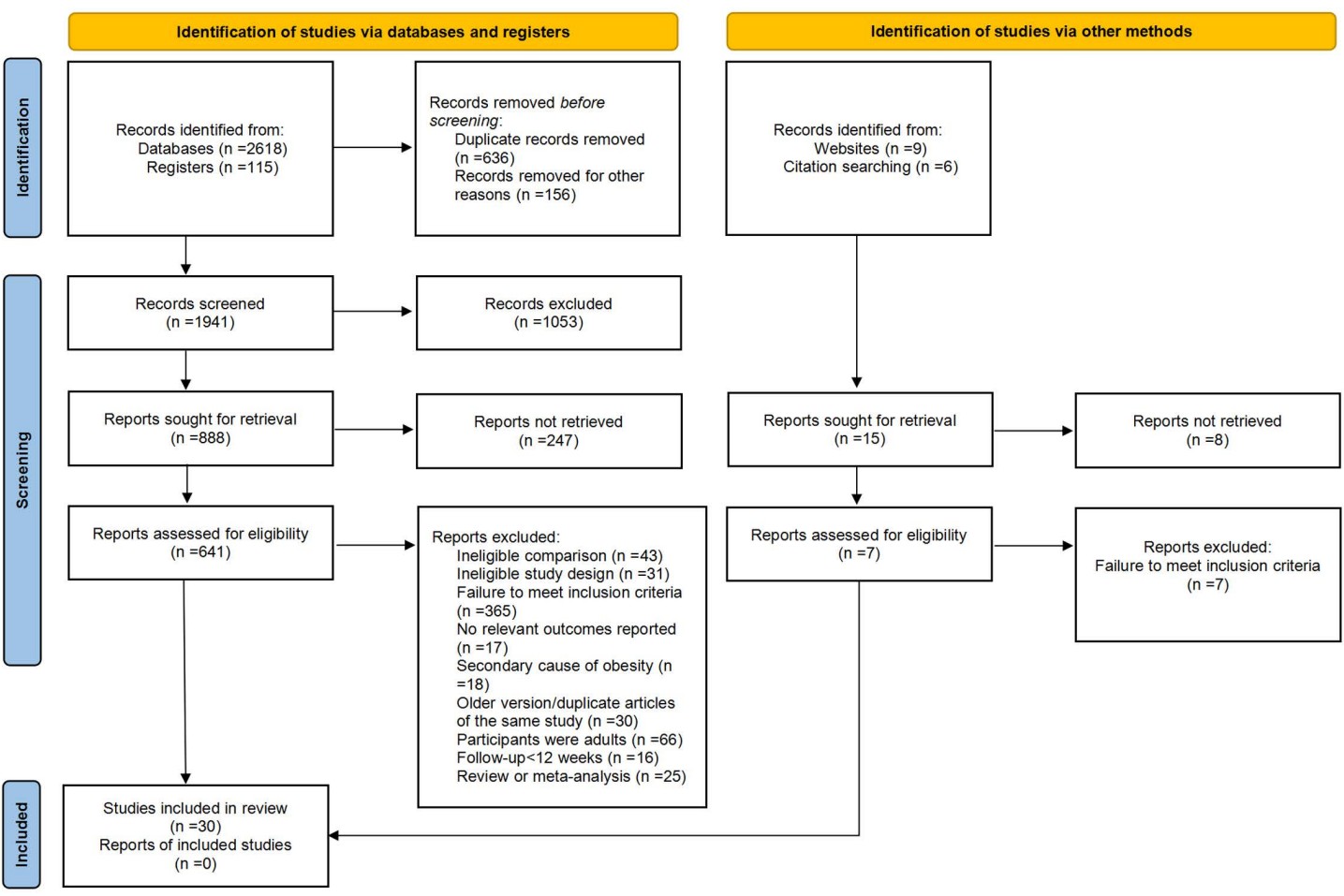

**Fig 1. Summary of study retrieval and identification for network meta-analysis.**

(Met/Flu), sibutramine, orlistat, and the control group (defined as placebo or no treatment). Sibutramine was pulled from the US market in 2010. All trials were two-group studies except Rezvanian 2010 [15], a multi-group study.

The years of publication it was ranged from 2003 to 2023. The mean duration of follow-up ranged between 12 and 68 weeks (median, 24; interquartile range, 24–47.75). The participants' mean age was 13.68 years [standard deviation (SD) 2.03], the mean baseline weight was 95.05 kg (SD 22.93), the mean height was 162.23 cm (SD 11.90), the mean BMI was 34.08 kg/m$^2$ (SD 6.14), and the mean waist circumference was 102.27 cm (SD 15.22). One study (223 participants) compared the PHEN/TPM to placebo, one trial (30 participants) compared the topiramate to placebo, and one trial (180 participants) was a multi-group study that compared metformin, fluoxetine, and Met/Flu. Among the five glucagon-like peptides 1 receptor agonists (GLP-1 RAs) with placebo trials, exenatide was the most commonly studied drug, with three trials (136 participants), one trial was semaglutide (201 participants), and one trial was liraglutide (251 participants). Thirteen trials (1265 participants) compared metformin with placebo or no treatment; four trials (655 participants) compared sibutramine with placebo; and five trials (881 participants) compared orlistat with placebo or no treatment. All studies included a background in lifestyle modification therapy except for Freemark 2007 [16], which has no behavior-changing intervention. Appendix 4 in S1 File displays a network of eligible comparisons for outcomes.

Two trials also included overweight participants (Maahs 2006 [17], Zahmatkesh 2023 [18]). In one trial (Weghuber 2022 [19]), only one participant (0.5%) in the semaglutide group was overweight. At baseline, a minimal number of participants in Weghuber 2022 had type 2 diabetes (4%). A person with dysglycemia had a fasting plasma glucose level of at least 100 mg per deciliter (≥5.6 mmol per liter), a glycated hemoglobin level of at least 5.7%, or both. This was found in 25.9% of people in Kelly in 2022 [20]. In Kelly 2020 [21], researchers included individuals with type 2 diabetes, and stratified blocked randomization was conducted based on blood glucose status (average blood glucose vs. pre-diabetes or type 2 diabetes), with mean fasting blood glucose at baseline of 94.3 mg/dl (SD 9.5). However, the proportion of people with type 2 diabetes is unclear in the two studies above. We contacted Professor Dr. Kelly, the lead author of the above two articles, to get additional study data but received no response.

### 3.2. Risk of bias of included trials

Appendix 5 in S1 File assesses the risk of bias in the included studies for our primary outcome. Of the 30 included studies, three (13%) [19,21–23] were deemed to have a low risk of bias across all domains. 17 (56.7%) reported adequate random sequence generation; 12 (40.0%) reported sufficient deviation from the intended interventions; 24 (80.0%) reported adequate measurement of the outcome; and 18 (60.0%) reported adequate selection of the reported result. 17 (56.74%) were at high risk of bias for missing outcome data. Overall, the risk of bias across the evidence network was relatively high.

At visual inspection, comparison-adjusted funnel plots for our primary outcome, BMI, appeared asymmetrical, as did the other outcome measures, DBP, HR, HOMA-IR, TC, and TG (Appendix 6 in S1 File).

### 3.3. Primary outcome: BMI and BMI percentage change

We included 29 trials (97%) involving 3224 participants (84%) in the network meta-analysis of BMI (Table 1). In comparison to the control groups, semaglutide (MD −5.90, 95% CI −8.13 to −3.65, moderate-certainty evidence), metformin (−0.85, −1.49 to −0.25, very low-certainty evidence), fluoxetine (−1.57, −2.82 to −0.35, very low-certainty evidence), Met/Flu combination therapy (−1.87, −3.12 to −0.65, very low-certainty evidence), PHEN/TPM combination therapy (−6.96, −10.44 to −3.43, very low-certainty evidence), orlistat (−1.38, −2.37 to −0.44, moderate-certainty evidence), and sibutramine (−1.37, −2.74 to −0.02, very low-certainty evidence) were all associated with a reduction in BMI. The efficacy of PHEN/TPM in reducing BMI was superior to that of exenatide (MD −5.62, 95% CI −9.42 to −1.79, very low-certainty evidence), liraglutide (−5.38, −9.45 to −1.28, moderate-certainty evidence), metformin (−6.11, −9.63 to −2.53, very low-certainty evidence), fluoxetine (−5.38, −9.08 to −1.65, low-certainty evidence), Met/Flu (−5.08, −8.76 to −1.34, very low-certainty evidence), topiramate (−6.09, −10.29 to −1.88, very low-certainty evidence), orlistat (−5.57, −9.18 to −1.92, very low-certainty evidence), and sibutramine (−5.59, −9.30 to −1.81, very low-certainty evidence). The study also revealed that semaglutide exhibited superior efficacy in reducing BMI compared to exenatide (MD −4.57, 95% CI −7.25 to −1.86, moderate-certainty evidence), liraglutide (−4.32, −7.39 to −1.24, moderate-certainty evidence), metformin (−5.05, −7.34 to −2.70, very low-certainty evidence), fluoxetine (−4.33, −6.86 to −1.76, very low-certainty evidence), Met/Flu (−4.02, −6.57 to −1.45, very low-certainty evidence), topiramate (−5.03, −8.28 to −1.79, very low-certainty evidence), orlistat (−4.52, −6.03 to −2.04, moderate-certainty evidence), and sibutramine (−4.53, −7.14 to −1.88, moderate-certainty evidence).

Nine trials (30%) involving 1273 participants (33% of the total) reported changes in BMI (Table 1). Semaglutide (MD −16.10, 95% CI −25.78 to −6.39, moderate-certainty evidence)

**Table 1. Network meta-analysis results for BMI and BMI percentage change.**

| | | | | | | | | | | |
|---|---|---|---|---|---|---|---|---|---|---|
| **Exenatide** | −0.99‡ (−12.48, 11.27) | **−12.43†** (**−23.95, −0.30**) | NA | NA | NA | 1.80‡ (−9.88, 13.92) | −6.04‡ (−17.64, 6.24) | NA | −2.19‡ (−10.41, 7.27) | 3.66† (−2.96, 10.93) |
| 0.24‡ (−2.36, 2.83) | **Liraglutide** | −11.48* (−25.06, 2.07) | NA | NA | NA | 2.76‡ (−11.03, 16.29) | −5.07‡ (−18.60, 8.60) | NA | −1.23‡ (−11.96, 10.07) | 4.63‡ (−4.93, 14.18) |
| **4.57*** (**1.86, 7.25**) | **4.32*** (**1.24, 7.39**) | **Semaglutide** | NA | NA | NA | 14.24† (0.39, 27.85) | 6.40‡ (−7.31, 20.15) | NA | 10.25* (−0.70, 21.59) | **16.10*** (**6.39, 25.78**) |
| −0.48‡ (−2.09, 1.14) | −0.73* (−2.92, 1.52) | **−5.05‡** (**−7.34, −2.70**) | **Metformin** | NA | NA | NA | NA | NA | NA | NA |
| 0.24‡ (−1.69, 2.18) | 0.00‡ (−2.45, 2.47) | **−4.33‡** (**−6.86, −1.76**) | 0.73‡ (−0.52, 1.95) | **Fluoxetine** | NA | NA | NA | NA | NA | NA |
| 0.55‡ (−1.39, 2.48) | 0.30‡ (−2.16, 2.78) | **−4.02‡** (**−6.57, −1.45**) | 1.03‡ (−0.22, 2.25) | 0.30‡ (−1.08, 1.68) | **Met/Flu** | NA | NA | NA | NA | NA |
| −0.47‡ (−3.25, 2.32) | −0.71‡ (−3.88, 2.47) | **−5.03‡** (**−8.28, −1.79**) | 0.02‡ (−2.44, 2.43) | −0.71‡ (−3.38, 1.93) | −1.01‡ (−3.70, 1.64) | **Topiramate** | −7.83† (−21.54, 6.01) | NA | −3.96‡ (−14.84, 7.52) | 1.88‡ (−7.81, 11.71) |
| **5.62‡** (**1.79, 9.42**) | **5.38*** (**1.28, 9.45**) | 1.06‡ (−3.09, 5.20) | **6.11‡** (**2.53, 9.63**) | **5.38†** (**1.65, 9.08**) | **5.08‡** (**1.34, 8.76**) | **6.09‡** (**1.88, 10.29**) | **PHEN/TPM** | NA | 3.84‡ (−7.06, 15.21) | 9.70† (−0.03, 19.30) |
| 0.05‡ (−1.72, 1.84) | −0.20‡ (−2.50, 2.17) | **−4.52*** (**−6.03, −2.04**) | 0.53‡ (−0.60, 1.68) | −0.19‡ (−1.74, 1.39) | −0.49‡ (−2.04, 1.09) | 0.52‡ (−2.01, 3.08) | **−5.57‡** (**−9.18, −1.92**) | **Orlistat** | NA | NA |
| 0.04‡ (−1.97, 2.05) | −0.21‡ (−2.71, 2.33) | **−4.53*** (**−7.14, −1.88**) | 0.52‡ (−0.97, 2.00) | −0.21‡ (−2.03, 1.62) | −0.51‡ (−2.33, 1.34) | 0.50‡ (−2.20, 3.23) | **−5.59‡** (**−9.30, −1.81**) | −1.01‡ (−1.69, 1.63) | **Sibutramine** | **5.86†** (**0.04, 11.12**) |
| −1.33‡ (−2.83, 0.15) | −1.57* (−3.70, 0.56) | **−5.90*** (**−8.13, −3.65**) | **−0.85‡** (**−1.49, −0.25**) | **−1.57‡** (**−2.82, −0.35**) | **−1.87‡** (**−3.12, −0.65**) | −0.87‡ (−3.21, 1.48) | **−6.96‡** (**−10.44, −3.43**) | **−1.38*** (**−2.37, −0.44**) | **−1.37‡** (**−2.74, −0.02**) | **Control** |

The mean difference (MD) with 95% confidence interval (CI) of network meta-analysis for BMI and change in BMI were listed in the left lower half (S1) and upper right half (S2).

S1. BMI ($N_T$ = 29 [97%], $n_t$ = 3224 [84%], $I^2$ = 0.0%).

S2. Change in BMI ($N_T$ = 9 [30%], $n_t$ = 1273 [33%], $I^2$ = 6.0%).

Comparisons should be read from left to right, and the estimate is in the cell in common between the column-defining treatment and the row-defining treatment. A MD of less than 0 indicates that the outcome is more likely with treatment (column) than reference (row). Significant results are in bold and underscored. The evidence is graded using the CINeMA system (Confidence in Network Meta-Analysis), an adaptation of the GRADE (Grading of Recommendations, Assessment, Development, and Evaluations) approach for network meta-analysis.

*BMI* body-mass index, *Met/Flu* metformin/ fluoxetine, *PHEN/TPM* phentermine/topiramate, *Control* placebo or no treatment, *NA* not available.

‡High quality of evidence.

*Moderate quality of evidence.

†Low quality of evidence.

‡Very low quality of evidence. $N_T$ total number of trials reporting the outcome (percentage of sample), $n_t$ total number of patients available for the respective outcome (percentage of sample).

and sibutramine (MD −5.86, 95% CI −11.12 to −0.04, low-certainty evidence) showed a significantly more significant reduction in BMI compared to the control groups. Regarding percentage change in BMI, semaglutide demonstrated superior efficacy over exenatide (−12.43, −23.95 to −0.30, low-certainty evidence).

## 3.4. Additional weight-related outcomes

Weight change (Table 2) was reported in 23 studies (77%) involving 2369 participants (62% of the total). Compared to control groups, semaglutide (MD −17.98, 95% CI −24.77 to −11.21), metformin (−2.76, −5.33 to −0.06), PHEN/TPM (−14.60, −21.06 to −8.11), orlistat (−5.32, −9.34 to −1.46), and sibutramine (−4.51, −7.58 to −1.17) were associated with significant reductions in weight. The study found that semaglutide effectively reduced weight compared to seven other AOMs, except for PHEN/TPM (MD ranging between −15.56 and −12.65).

**Table 2. Results of a network meta-analysis for weight and BMI-SDS.**

| Exenatide | −0.13† (−0.44, 0.18) | NA | 0.00† (−0.28, 0.28) | NA | NA | NA | −0.03‡ (−0.34, 0.28) | 0.10† (−0.17, 0.38) |
|---|---|---|---|---|---|---|---|---|
| 0.24‡ (−7.68, 8.07) | Liraglutide | NA | 0.13† (−0.02, 0.28) | NA | NA | NA | 0.10‡ (−0.10, 0.30) | **0.23‡ (0.09, 0.37)** |
| **13.72* (5.50, 21.89)** | **13.47* (4.13, 22.79)** | Semaglutide | NA | NA | NA | NA | NA | NA |
| −1.51‡ (−6.92, 3.67) | −1.74‡ (−8.74, 5.09) | **−15.21* (−22.56, −8.00)** | Metformin | NA | NA | NA | −0.03‡ (−0.18, 0.12) | **0.10* (0.04, 0.16)** |
| −1.85‡ (−10.19, 6.41) | −2.07‡ (−11.50, 7.43) | **−15.56† (−25.22, −5.88)** | −0.33‡ (−7.72, 7.10) | Topiramate | NA | NA | NA | NA |
| **10.31† (2.33, 18.21)** | **10.08* (1.01, 19.19)** | −3.39‡ (−12.79, 5.99) | **11.83† (4.89, 18.85)** | **12.17† (2.68, 21.63)** | PHEN/TPM | NA | NA | NA |
| 1.05‡ (−5.00, 7.14) | 0.81‡ (−6.58, 8.42) | **−12.65* (−20.47, −4.77)** | 2.56‡ (−2.05, 7.47) | 2.88‡ (−5.02, 10.95) | **−9.27† (−16.74, −1.59)** | Orlistat | NA | NA |
| 0.23‡ (−5.54, 5.66) | 0.01‡ (−7.27, 7.01) | **−13.47* (−21.08, −6.12)** | 1.74‡ (−2.43, 5.83) | 2.08‡ (−5.65, 9.57) | **−10.08† (−17.40, −2.98)** | −0.81‡ (−6.12, 4.05) | Sibutramine | 0.13‡ (−0.01, 0.27) |
| −4.27‡ (−8.92, 0.26) | −4.51† (−10.92, 1.91) | **−17.98* (−24.77, −11.21)** | **−2.76† (−5.33, −0.06)** | −2.43‡ (−9.36, 4.51) | **−14.60† (−21.06, −8.11)** | **−5.32* (−9.34, −1.46)** | **−4.51† (−7.58, −1.17)** | Control |

The mean difference (MD) with 95% confidence interval (CI) of network meta-analysis for weight and BMI-SDS were listed in the left lower half (S1) and upper right half (S2).

S1. Weight ($N_T$ = 23 [77%], $n_t$ = 2369 [62%], $I^2$ = 0.0%).

S2. BMI-SDS ($N_T$ = 7 [23%], $n_t$ = 741 [19%], Bayesian fixed-effects model, $I^2$ = 0.0%).

Comparisons should be read from left to right, and the estimate is in the cell in common between the column-defining treatment and the row-defining treatment. A MD of less than 0 indicates that the outcome is more likely with treatment (column) than reference (row). Significant results are in bold and underscored. The evidence is graded using the CINeMA system (Confidence in Network Meta-Analysis), an adaptation of the GRADE (Grading of Recommendations, Assessment, Development, and Evaluations) approach for network meta-analysis.

*BMI* body-mass index, *BMI-SDS* body mass index-SD score, *PHEN/TPM* phentermine/topiramate, *Control* placebo or no treatment, *NA* not available.

‡High quality of evidence.

*Moderate quality of evidence.

†Low quality of evidence.

‡Very low quality of evidence. $N_T$ total number of trials reporting the outcome (percentage of sample), $n_t$ total number of patients available for the respective outcome (percentage of sample).

Similarly, PHEN/TPM had greater efficacy in weight reduction than seven other AOMs, except semaglutide (MD ranging between −12.17 and −9.27).

Out of the total number of studies, 7 (23%) involving 741 participants (19% of the total) reported changes in BMI-SDS (Table 2). Only four interventions—exenatide, liraglutide, metformin, and sibutramine—yielded outcomes for BMI-SDS. Liraglutide and metformin exhibited a more significant reduction in BMI-SDS compared to the control groups (−0.23, −0.37 to −0.09, and −0.10, −0.16 to −0.04, respectively).

Change in waist circumference data was provided in 14 (47%) trials with 1755 (46%) participants (Table 3). It is worth noting that only the exenatide group did not exhibit significant changes compared to the control groups. Semaglutide has shown superior efficacy in decreasing waist circumference compared to eight other AOMs, except PHEN/TPM (MD ranging from −11.61 to −6.07). The PHEN/TPM significantly reduced waist circumference compared to eight other AOMs, except for semaglutide and sibutramine (MD ranging between −8.64 and −5.51). Sibutramine had superior efficacy compared to exenatide, metformin, fluoxetine, Met/Flu, and orlistat, with MD ranging from −5.54 to −2.46. Orlistat has shown superior efficacy compared to metformin (−3.07, −4.43 to −1.78), fluoxetine (−3.08, −4.43 to −1.78), and Met/Flu (−2.38, −3.73 to −1.09). The Met/Flu demonstrated greater

**Table 3. Network meta-analysis results for waist circumference.**

| Exenatide | | | | | | | | | |
|---|---|---|---|---|---|---|---|---|---|
| 1.61‡ (−3.03, 6.26) | Liraglutide | | | | | | | | |
| **10.43*** **(6.15, 14.30)** | **8.88*** **(5.56, 12.02)** | Semaglutide | | | | | | | |
| −1.50‡ (−4.77, 2.13) | **−2.71*** **(−5.24, −0.56)** | **−11.60†** **(−13.91, −9.41)** | Metformin | | | | | | |
| −1.51‡ (−4.77, 2.13) | **−2.72*** **(−5.25, −0.57)** | **−11.61†** **(−13.91, −9.41)** | −0.00‡ (−0.09, 0.06) | Fluoxetine | | | | | |
| −0.81‡ (−4.07, 2.84) | −2.01‡ (−4.55, 0.13) | **−10.91†** **(−13.22, −8.72)** | **0.70†** **(0.60, 0.78)** | **0.70†** **(0.61, 0.79)** | Met/Flu | | | | |
| **7.25†** **(2.22, 11.44)** | **5.86*** **(2.09, 9.58)** | −2.88‡ (−6.63, 0.44) | **8.64†** **(6.01, 11.51)** | **8.64†** **(6.01, 11.50)** | **7.94†** **(5.31, 10.81)** | PHEN/TPM | | | |
| 1.70‡ (−2.01, 5.77) | 0.35‡ (−2.31, 2.94) | **−8.54*** **(−11.18, −5.97)** | **3.07†** **(1.78, 4.43)** | **3.08†** **(1.78, 4.43)** | **2.38†** **(1.09, 3.73)** | **−5.51†** **(−8.86, −2.61)** | Orlistat | | |
| **4.13†** **(1.03, 8.23)** | 2.80* (−0.12, 5.68) | **−6.07*** **(−8.84, −2.95)** | **5.54†** **(4.22, 7.35)** | **5.54†** **(4.22, 7.35)** | **4.84†** **(3.52, 6.65)** | −3.07‡ (−6.21, 0.42) | **2.46*** **(0.40, 4.67)** | Sibutramine | |
| −1.71‡ (−4.97, 1.93) | **−2.92#** **(−5.45, −0.78)** | **−11.81*** **(−14.11, −9.62)** | **−0.20†** **(−0.32, −0.15)** | **−0.20†** **(−0.29, −0.14)** | **−0.90†** **(−1.00, −0.82)** | **−8.85†** **(−11.71, −6.22)** | **−3.28*** **(−4.63, −1.99)** | **−−5.75†** **(−7.55, −4.43)** | Control |

The mean difference (MD) with 95% confidence interval (CI) of network meta-analysis for waist circumference.

Waist circumference ($N_T$ = 14 [47%], $n_t$ = 1755 [46%], $I^2$ = 5.0%).

Comparisons should be read from left to right, and the estimate is in the cell in common between the column-defining treatment and the row-defining treatment. A MD of less than 0 indicates that the outcome is more likely with treatment (column) than reference (row). Significant results are in bold and underscored. The evidence is graded using the CINeMA system (Confidence in Network Meta-Analysis), an adaptation of the GRADE (Grading of Recommendations, Assessment, Development, and Evaluations) approach for network meta-analysis.

*Met/Flu* metformin/ fluoxetine, *PHEN/TPM* phentermine/topiramate, *Control* placebo or no treatment, *NA* not available.

#High quality of evidence.

*Moderate quality of evidence.

†Low quality of evidence.

‡Very low quality of evidence. $N_T$ total number of trials reporting the outcome (percentage of sample), $n_t$ total number of patients available for the respective outcome (percentage of sample).

efficacy than metformin and fluoxetine (−0.70, −0.78 to −0.60 and −0.70, −0.79 to −0.61, respectively).

## 3.5. Metabolic outcomes

14 (47%) trials involving 1458 (38%) participants provided changes in FBG data (Appendix 8 in S1 File). Compared with the control groups, semaglutide (MD −0.20, 95% CI −0.36 to −0.04) and orlistat (−0.24, −0.38 to −0.10) were associated with lowering FBG. Orlistat exhibited a more significant reduction in FBG than metformin (−0.19, −0.35 to −0.03). 17 (57%) trials with 1329 (35%) participants provided changes in FINS data. Metformin showed a more significant reduction in FINS (Appendix 8 in S1 File) than the control groups (−3.81, −7.98 to −0.24). 13 (43%) trials with 1244 (33%) participants provided changes in HOMA-IR data (Appendix 8 in S1 File). Sibutramine demonstrated a more remarkable significant improvement in HOMA-IR compared to metformin (−11.51, −18.20 to −4.80), orlistat (−11.43, −18.65 to −4.15), and the control groups (−12.22, −18.82 to −5.59).

Out of the total number of studies, 16 studies (53%) with 1373 participants (36%) provided results about the change in HDL-C (Appendix 8 in S1 File); 16 studies (53%) provided results about the change in LDL-C (Appendix 8 in S1 File), with a total of 1091 (29%) individuals; and

17 studies (57%) with 1427 participants (37%) provided results about the change in TG (Appendix 8 in S1 File). The network meta-analysis found no statistically significant difference between the medication classes in the above-mentioned outcomes. However, in terms of improving TC (Appendix 8 in S1 File), 14 studies (47%) with a total of 1012 people (26%) reported data on changes. We found that orlistat was more effective than metformin (−6.64, −13.17 to −0.16), sibutramine (−26.35, −46.73 to −6.03), and the control groups (−8.77, −13.96 to −3.57).

### 3.6. Anthropometric outcomes

Sibutramine elevated systolic blood pressure (SBP) (Appendix 8 in S1 File) compared to exenatide (MD 4.53, 95% CI 0.68 to 8.39), liraglutide (3.16, 0.53 to 5.79), metformin (1.45, 0.44 to 2.44), and the control groups (1.12, 0.24 to 1.99). There were no significant differences in diastolic blood pressure (DBP) (Appendix 8 in S1 File) across different drug classes. Compared with metformin and the control groups, sibutramine increased heart rate (12.78, 1.32 to 25.24 and 4.71, respectively) (Appendix 8 in S1 File).

### 3.7. Safety outcomes

The network meta-analysis found no significant difference between drug classes in the treatment of gastrointestinal disorders, depression, and serious adverse events (Appendix 8 in S1 File).

### 3.8. Drug class rankings

Appendix 9 in S1 File shows the mean values of SUCRA for providing the hierarchy ranking of different treatments on our outcomes. The PHEN/TPM was likely to rank best for lower BMI, DBP, and gastrointestinal disorder outcomes. Research demonstrated that semaglutide was the most effective in reducing BMI percentage change, weight, and waist circumference outcomes. Liraglutide was most likely to rank best for decreased BMI-SDS. For a lower SBP, exenatide was possible to rank the best. We found that metformin was the most effective for lowering HR. Topiramate was the most effective in lowering FBG, TG, and SAE. Sibutramine was most likely to rank best for lower FINS, HOMA-IR, and HDL-C. Orlistat was most likely to rank best for decreased TC and LDL-C. In terms of safety results, semaglutide may be associated with a higher incidence of gastrointestinal disorders. The combination therapy of PHEN/TPM has the potential to result in heightened depression and significant adverse events.

### 3.9. Subgroup analysis, sensitivity analysis, and meta-regression analysis

Subgroup analysis (Appendix 10 in S1 File) revealed that metformin and sibutramine significantly reduced BMI in the group with a BMI ≥ 35 kg/m² compared to the control groups. Conversely, orlistat showed more significant benefits than the control groups in the BMI < 35 kg/m² group.

We performed the sensitivity analysis (Appendix 11 in S1 File) by excluding three open-label trials [24–26]. The sensitivity analyses did not affect the associations of some AOMs with reduced BMI outcomes, except orlistat with the control group.

According to the network meta-regression analysis (Appendix 12 in S1 File), covariates didn't affect outcomes. They only did this when comparing orlistat to controls, where baseline BMI did affect BMI.

### 3.10. Transitivity, heterogeneity, and quality of the evidence

We assumed the transitivity assumption was valid because we specifically selected specific populations and only used indirect estimates from placebo or no treatment nodes in our

network meta-analysis. Heterogeneity (global $I^2$) was low for our outcomes (range, 0.0% to 13.0%, Tables 1–3, and Appendix 8 in S1 File).

Concerning the primary outcomes, we judged the confidence in the evidence for 17.1% of the 76 comparisons to be high or moderate, 9.2% to be low, and 73.7% to be very low. For the additional weight-related outcomes, 24.2% of the 91 comparisons were high or moderate, 39.6% were low, and 36.3% were very low. Regarding metabolic outcomes, 3.8% of the 208 comparisons were high or moderate, 3.4% were low, and 44.2% were very low. Due to the lack of closing loops, we only downgraded the confidence of evidence in network meta-analysis results due to within-study bias, imprecision, and reporting bias (Appendix 13 in S1 File).

## 4. Discussion

This network meta-analysis provides a comprehensive summary of the efficacy of available drug treatments for children and adolescents with obesity, including reductions in body weight and improvements in cardiometabolic risk factors (waist circumference and metabolic outcomes). We included 30 randomized clinical trials, enrolling 3822 participants. Semaglutide and PHEN/TPM therapy were associated with significant reductions in BMI, BMI percentage change, and additional weight-related outcomes. In terms of safety outcomes, our review suggests that semaglutide may be associated with more gastrointestinal disorders. PHEN/TPM combination therapy could potentially lead to increased depression and serious adverse events in terms of safety outcomes. Overall, semaglutide may be preferred over the eleven treatments based on their statistically significant reductions in body weight.

To our knowledge, this study is the first comprehensive systematic review and network meta-analysis assessing all pertinent drug classes for children and adolescents with obesity. Furthermore, due to their ongoing growth, we typically consider age and sex when assessing a child's weight status. So, in this review, we also use BMI-SDS as an alternative efficacy measure [27]. However, only seven trials reported changes in BMI-SDS. We also faced problems in our meta-analysis as some trials did not report the raw data we required, so we had to try and obtain this from the trial authors. We analyzed baseline BMI, age, and length of follow-up on our primary BMI outcomes and performed meta-regression analysis, resulting in small changes in point estimates.

In support of the Endocrine Society's pediatric obesity treatment clinical practice guideline [28], a systematic review [29] concluded that a BMI reduction of at least 1.6 kg/m2 could be considered clinically meaningful. Previous meta-analyses [30] found that pharmacological interventions (metformin, sibutramine, orlistat, and fluoxetine) may have minor effects on a reduction in BMI (MD −1.3, 95% CI −1.9 to −0.8) and body weight (MD −3.9, 95% CI −5.9 to −1.9) in obese children and adolescents. Another meta-analysis [31] involving three exenatide and six liraglutide studies evaluated GLP-1RAs in children and adolescents with obesity and found that their use led to a slight reduction in body weight (MD −1.50, 95% CI −2.50 to −0.50) and BMI (MD −1.24, 95% CI −1.71 to −0.77). Furthermore, some studies on liraglutide had a short observation period and included patients with type 2 diabetes, which interfered with weight loss maintenance and metabolic markers, especially glycemic markers. Our study included more recent trials and more interventions (PHEN/TPM, Semaglutide, Exenatide, Liraglutide, and Met/Flu), in which PHEN/TPM and semaglutide showed significant efficacy in reducing body weight; semaglutide significantly reduced body weight and BMI compared to other GLP-1RAs like exenatide and liraglutide.

Children rarely experience cardiovascular events [32], but studies have identified BMI, triglycerides, and total cholesterol levels as childhood risk factors linked to subsequent cardiovascular events in adulthood [33]. Meta-analyzing metabolic outcomes were complex because

some trials didn't give us the raw data we needed, and we couldn't get raw data from the trial authors either. This means that we could not correctly compare the effects of semaglutide and PHEN/TPM on metabolic markers in our evidence network. Semaglutide use was associated with improvements in cardiometabolic endpoints such as waist circumference, levels of glycated hemoglobin, and lipids (excluding high-density lipoprotein cholesterol) in randomized controlled research [19]. The PHEN/TPM study [20] found that using PHEN/TPM decreased triglyceride levels and increased HDL-C levels. Despite not reaching statistical significance after accounting for numerous secondary endpoints, the size of these alterations was comparable to previous findings in adults [34,35].

Anti-obesity therapies' tolerability is critical for promoting adherence and subsequent effectiveness [36,37]. Although semaglutide was associated with more gastrointestinal disorders than any other treatment in our study, permanent discontinuations due to gastrointestinal disorders were very low. Among all the treatments in our study, PHEN/TPM combination therapy may lead to increased depression and SAE. Two participants in the top-dose group reported SAE [20], and the safety profile of PHEN/TPM was consistent with the findings from adult trials [34,35,38].

## 5. Strengths and limitations

This study is the first comprehensive systematic review and network meta-analysis assessing all pharmacological classes for pediatric and adolescent populations with obesity, offering a synthesis of available therapeutic options, including weight reduction strategies and enhancements in cardiometabolic risk factors. However, our study has some limitations, which are worth further discussion. First, there were no head-to-head comparisons of pharmacological therapies in trials for weight and metabolic outcomes in pediatric and adolescent obesity. We generated the comparative effects using indirect evidence. We only found placebo-controlled studies, so none of the networks had closed loops in our primary outcomes. We acknowledge the limitations of our current network meta-analysis, specifically the need for mixed evidence. Secondly, the study's low test efficiency stems from the small number of uncovered trials. Third, the study restricted itself to English, potentially leading to language bias. Fourth, the CINeMA framework rated some comparisons as low or very low quality, potentially limiting the interpretation of the results. Lastly, our meta-analysis could not conduct an individual patient data meta-analysis to investigate potentially significant impact modifiers in our study comprehensively.

## 6. Future of AOM for the pediatric population and research gaps

The third generation of nutrient-stimulated hormone-based (NuSH) anti-obesity medications (AOM), which includes co- and tri-agonists that combine peptides such as GLP-1, glucose-dependent insulinotropic polypeptide (GIP), amylin, and glucagon, is poised to commence clinical trials in adolescents. The inaugural trial involving tripeptide is anticipated to initiate in mid-2024. Subsequent studies should address limitations, including long-term safety, durability of treatment response, and the inclusion of broader populations. The distinctive biopsychosocial context experienced by children and adolescents during cognitive, psychological, and pubertal development is essential for ensuring long-term safety. Future research should evaluate the real-world adoption of weight-loss pharmacotherapies among adolescents, their effects on metabolic health, and potential strategies to mitigate weight regain upon discontinuation. Clinicians are pivotal in educating patients regarding lifestyle modifications, pharmacotherapy options, and surgical interventions aimed at preventing adult obesity. Furthermore, future investigations should expand eligibility criteria to represent the demographics utilizing AOM accurately.

## 7. Conclusions

In this network meta-analysis, in children and adolescents with obesity, the use of semaglutide outperformed various AOMs in reducing BMI, while PHEN/TPM showed comparable efficacy. Semaglutide demonstrated comparable effectiveness in the percentage change in BMI compared to exenatide, placebo, or no treatment. Semaglutide and PHEN/TPM also exhibited superior efficacy in reducing waist circumference compared to most AOMs. However, the overall quality of comparative evidence is not high. Therefore, more large-scale, longer-term, well-organized, multi-center, double-blind, placebo-controlled studies with rigorous designs and extended post-interventional monitoring are needed to provide more objective clinical data for guiding clinical treatment.

## Supporting information

**S1 File. Supplementary Appendix: Methodological Details and Results of Network Meta-Analysis.**
(DOC)

**S1 Checklist. PRISMA 2020 checklist.**
(DOCX)

## Author contributions

**Conceptualization:** Peizhuo Zang.

**Data curation:** Shuo Yang, Shuangqing Xin.

**Formal analysis:** Shuo Yang, Ronghui Ju.

**Investigation:** Shuangqing Xin.

**Methodology:** Ronghui Ju.

**Validation:** Ronghui Ju.

**Writing – original draft:** Shuo Yang.

**Writing – review & editing:** Peizhuo Zang.

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
