## [Decision Letter · Decision Letter 0]

7 Oct 2024

PONE-D-24-40184Pharmacological Interventions for Addressing Pediatric and Adolescent Obesity: A Systematic Review and Network Meta-analysisPLOS ONE

Dear Dr. Yang,

Thank you for submitting your manuscript to PLOS ONE. After careful consideration, we feel that it has merit but does not fully meet PLOS ONE’s publication criteria as it currently stands. Therefore, we invite you to submit a revised version of the manuscript that addresses the points raised during the review process.

We look forward to receiving your revised manuscript.

Kind regards,

Aleksandra Klisic

Academic Editor

PLOS ONE

3. As required by our policy on Data Availability, please ensure your manuscript or supplementary information includes the following:

Reviewers' comments:

Reviewer's Responses to Questions

**Comments to the Author**

1. Is the manuscript technically sound, and do the data support the conclusions?

Reviewer #1: Partly

Reviewer #2: Yes

2. Has the statistical analysis been performed appropriately and rigorously? 

Reviewer #1: Yes

Reviewer #2: Yes

3. Have the authors made all data underlying the findings in their manuscript fully available?

Reviewer #1: No

Reviewer #2: Yes

4. Is the manuscript presented in an intelligible fashion and written in standard English?

Reviewer #1: Yes

Reviewer #2: Yes

5. Review Comments to the Author

Reviewer #1: Yang et al. have performed a systematic review and network meta-analysis of pharmacologic interventions for pediatric and adolescent obesity. These are my comments:

- The abstract section is so long in its current format. It should be reduced to a maximum of 250-300 words.

- Authors should add strengths to their manuscript before the limitation.

- A paragraph highlighting the future research perspective and possible gaps in the literature should be added before the conclusion section.

- The references prior to 2010 could be updated with those after 2010 since they provide more up-to-date findings.

Reviewer #2: The manuscript titled "Pharmacological Interventions for Addressing Pediatric and Adolescent Obesity: A Systematic Review and Network Meta-analysis" evaluated the effects of medications on obesity in pediatric and adolescent. The manuscript is well-written with fair methodology. I have some comments for improvement:

1- Define abbreviations in their first use and make sure that abbreviated forms are being used after the definition.

2- I found some typos and grammatical errors. A native review is required.

3- Add the strengths of your study before the limitations paragraph.

4- The tables will be published without colors, Update them in the revised version.

6. PLOS authors have the option to publish the peer review history of their article (what does this mean? ). If published, this will include your full peer review and any attached files.

**Do you want your identity to be public for this peer review?** For information about this choice, including consent withdrawal, please see our Privacy Policy .

Reviewer #1: No

Reviewer #2: No

---

## [Author Response · Author response to Decision Letter 1]

30 Oct 2024

Dear Editors and Reviewers:

Thank you for your letter and for the reviewers’ comments concerning our manuscript entitled “Pharmacological Interventions for Addressing Pediatric and Adolescent Obesity: A Systematic Review and Network Meta-analysis.” (ID: PONE-D-24-40184). Those comments are all valuable and very helpful for revising and improving our paper, as well as providing important guiding significance for our research. We have studied the comments carefully and have made corrections, which we hope will be met with approval. The revised portions are marked in red on the paper. The main corrections in the paper and the response to the reviewer’s comments are as follows:

Responds to the editor's comments:

In accordance with the requirements of the journal's data availability policy, we integrated the features of excluded references and the risk of bias assessment for each included study into the supplementary information of the article. For detailed information, please refer to Appendix 4 Supplemental Table 4 and Appendix 6 Supplemental Table 6.

A special thanks to you for your good comments.

Responds to the reviewer’s comments:

Reviewer #1:

We are very grateful for your professional comments on the methodology of this revised manuscript. According to your advice, we amended the relevant part of the manuscript. All of your questions were answered one-by-one.

1.Response to comment: (The abstract section is so long in its current format. It should be reduced to a maximum of 250-300 words.）

Response: Thanks for the constructive suggestions to improve our manuscript. We have made the following changes to the abstract:

Background Obesity significantly impacts the health outcomes of children and adolescents, necessitating a comprehensive study to evaluate the effects of various anti-obesity medications (AOMs) on weight-related and metabolic outcomes.

Methods PubMed, EMBASE, and CENTRAL were searched for studies published up to January 3, 2024. We performed a network meta-analysis on randomized clinical trials that compared various treatments for pediatric and adolescent obesity, such as phentermine/topiramate, semaglutide, exenatide, liraglutide, topiramate, metformin, fluoxetine, metformin/fluoxetine, sibutramine, and orlistat. The study evaluated body mass index (BMI), BMI percentage change, weight, BMI-SDS, waist circumference, metabolic, anthropometric, and safety outcomes.

Results The study gathered 2733 studies, including 30 articles that involved 3822 participants. The results of our research showed that PHEN/TPM was better at lowering BMI than exenatide, liraglutide, metformin, fluoxetine, Met/Flu, topiramate, orlistat, and sibutramine, with mean differences (MD) ranging from -10.29 to -1.28. Additionally, semaglutide demonstrated superior efficacy over other AOMs (MD ranged from -8.28 to -1.24). Various levels of certainty, ranging from very low to moderate, supported the findings. Furthermore, semaglutide demonstrated superior efficacy over exenatide (MD-12.43, 95% CI -23.95 to -0.30) regarding percentage change in BMI. Semaglutide also showed enhanced weight reduction effectiveness compared to seven other AOMs except for PHEN/TPM (MD ranging from -15.56 to -12.65). Similarly, PHEN/TPM displayed greater weight reduction effectiveness than seven other AOMs, except for semaglutide (MD ranged from -12.17 to -9.27). Moreover, semaglutide proved more effective in decreasing waist circumference when compared with other AOMs apart from PHEN/TPM (MD ranged from -11.61 to -6.07). Similarly, we found that PHEN/TPM, excluding semaglutide and sibutramine, was more effective in reducing waist circumference (MD ranged from -8.64 to -5.51).

Conclusions The study found that semaglutide outperformed other AOMs in reducing BMI and additional weight-related outcomes in children and adolescents with obesity, while PHEN/TPM showed comparable efficacy.

2.Authors should add strengths to their manuscript before the limitation.

Response: We have enhanced the manuscript with the following strengths: This study is the first comprehensive systematic review and network meta-analysis assessing all pharmacological classes for pediatric and adolescent populations with obesity, offering a synthesis of available therapeutic options, including weight reduction strategies and enhancements in cardiometabolic risk factors. However, our study has some limitations, which are worth further discussion.

3.A paragraph highlighting the future research perspective and possible gaps in the literature should be added before the conclusion section.

Response: What we have added is as follows：

Future of AOM for the pediatric population and research gaps

The third generation of nutrient-stimulated hormone-based (NuSH) anti-obesity medications (AOM), which includes co- and tri-agonists that combine peptides such as GLP-1, glucose-dependent insulinotropic polypeptide (GIP), amylin, and glucagon, is poised to commence clinical trials in adolescents. The inaugural trial involving tirzepatide is anticipated to initiate in mid-2024. Subsequent studies should address limitations including long-term safety, durability of treatment response, and the inclusion of broader populations. The distinctive biopsychosocial context experienced by children and adolescents during cognitive, psychological, and pubertal development is essential for ensuring long-term safety. Future research should evaluate the real-world adoption of weight-loss pharmacotherapies among adolescents, their effects on metabolic health, and potential strategies to mitigate weight regain upon discontinuation. Clinicians are pivotal in educating patients regarding lifestyle modifications, pharmacotherapy options, and surgical interventions aimed at preventing adult obesity. Furthermore, future investigations should expand eligibility criteria to more accurately represent the demographics utilizing AOM.

4.The references prior to 2010 could be updated with those after 2010 since they provide more up-to-date findings.

Response: We have updated the references prior to 2010 as follows：

Reference 5. Singh AS, Mulder C, Twisk JW, van Mechelen W, Chinapaw MJ. Tracking of childhood overweight into adulthood: a systematic review of the literature. Obes Rev. 2008;9(5):474-88.

updated to: Chamay Weber C, Gal-Duding C, Maggio AB. Family based behavioral treatment in adolescents suffering from obesity: evolution through adulthood. BMC Pediatr. 2024;24(1):33.

However, references 15, 16, 17, 24 and 26 were the original research references for this meta-analysis and we did not replace them.

Special thanks to you for your good comments.

Reviewer #2:

1.Define abbreviations in their first use and make sure that abbreviated forms are being used after the definition.

Response: We have revised the use of abbreviations in accordance with your suggestion.

2.A review of typos and grammar is proposed.

Response: We have checked the article for typos and grammar according to your request.

3.Add the strengths of your study before the limitations paragraph.

Response: We have enhanced the manuscript with the following strengths: This study is the first comprehensive systematic review and network meta-analysis assessing all pharmacological classes for pediatric and adolescent populations with obesity, offering a synthesis of available therapeutic options, including weight reduction strategies and enhancements in cardiometabolic risk factors.

4.The tables will be published without colors, Update them in the revised version.

Response: We have changed the color of the table.

We tried our best to improve the manuscript and made some changes to it. These changes will not influence the content and framework of the paper. We appreciate the editors’ and reviewers' hard work earnestly and hope that the correction will be met with approval.

Once again, thank you very much for your comments and suggestions.

---

## [Decision Letter · Decision Letter 1]

18 Nov 2024

Pharmacological Interventions for Addressing Pediatric and Adolescent Obesity: A Systematic Review and Network Meta-analysis

PONE-D-24-40184R1

Dear Dr. Yang,

We’re pleased to inform you that your manuscript has been judged scientifically suitable for publication and will be formally accepted for publication once it meets all outstanding technical requirements.

Kind regards,

Aleksandra Klisic

Academic Editor

PLOS ONE

Additional Editor Comments (optional):

Reviewers' comments:

Reviewer's Responses to Questions

**Comments to the Author**

1. If the authors have adequately addressed your comments raised in a previous round of review and you feel that this manuscript is now acceptable for publication, you may indicate that here to bypass the “Comments to the Author” section, enter your conflict of interest statement in the “Confidential to Editor” section, and submit your "Accept" recommendation.

Reviewer #1: All comments have been addressed

Reviewer #2: All comments have been addressed

2. Is the manuscript technically sound, and do the data support the conclusions?

Reviewer #1: (No Response)

Reviewer #2: (No Response)

3. Has the statistical analysis been performed appropriately and rigorously? 

Reviewer #1: (No Response)

Reviewer #2: (No Response)

4. Have the authors made all data underlying the findings in their manuscript fully available?

Reviewer #1: (No Response)

Reviewer #2: (No Response)

5. Is the manuscript presented in an intelligible fashion and written in standard English?

Reviewer #1: (No Response)

Reviewer #2: (No Response)

6. Review Comments to the Author

Reviewer #1: (No Response)

Reviewer #2: (No Response)

7. PLOS authors have the option to publish the peer review history of their article (what does this mean? ). If published, this will include your full peer review and any attached files.

**Do you want your identity to be public for this peer review?** For information about this choice, including consent withdrawal, please see our Privacy Policy .

Reviewer #1: No

Reviewer #2: No

---

## [Editor Report · Acceptance letter]

PONE-D-24-40184R1

PLOS ONE

Dear Dr. Yang,

I'm pleased to inform you that your manuscript has been deemed suitable for publication in PLOS ONE. Congratulations! Your manuscript is now being handed over to our production team.

Kind regards,

on behalf of

Dr. Aleksandra Klisic

Academic Editor

PLOS ONE